# Changes in Consumers’ Food Purchase and Transport Behaviors over a Decade (2010 to 2019) Following Health and Convenience Food Trends

**DOI:** 10.3390/ijerph17155448

**Published:** 2020-07-29

**Authors:** Tae Jin Cho, Sun Ae Kim, Hye Won Kim, Sun Min Park, Min Suk Rhee

**Affiliations:** 1Department of Food and Biotechnology, College of Science and Technology, Korea University, Sejong 30019, Korea; microcho@korea.ac.kr; 2Department of Food Science and Engineering, Ewha Womans University, Seoul 03760, Korea; sunaekim@ewha.ac.kr; 3Department of Biotechnology, College of Life Sciences and Biotechnology, Korea University, Seoul 02841, Korea; kgpdnjs@korea.ac.kr (H.W.K.); ash101@korea.ac.kr (S.M.P.)

**Keywords:** consumer survey, food trend, food preparation behavior, food purchase/transport time, risk perception, healthy food consumption, cultural consumer context, food safety, convenience, microbiological risk

## Abstract

Although consumers’ food purchase/transport have been reported as causes of food safety risks, there is a lack of empirical data that are feasible to identify persistent and emerging risky behaviors of consumers. This longitudinal trend study consists of individual consumer surveys in 2010 (*n* = 609) and 2019 (*n* = 605) to analyze changes in risky behaviors linked to food purchase/transport over a decade. Overall, the results identified purchase/transport time and purchase order as the emerging and unchanged risk factors, respectively. Consumers’ preferences into channels for purchase (large discount stores rather than small/traditional markets) and transport (using cars or delivery) implied the convenience as the noticeable trend. Whereas, unexpected increases in purchase/transport time highlighted the underestimated risks in long-term exposure of foods under inadequate temperature. Food should not be exposed to danger zones > 1–2 h, but consumers might be unaware of the risk especially for preferred channels (e.g., 77 and 36 min. are required for purchase and transport from large discount stores, respectively). In the case of unchanged risky behavior, more than half of consumers in both surveys did not follow proper purchasing orders. Our findings highlight the necessity for novel countermeasures and the improvement of current consumer guidelines against emerging and unchanged risky behaviors, respectively.

## 1. Introduction

Consumers’ food and meal preparation behaviors from shopping to consumption have been associated with various human health issues, including foodborne diseases [1,2,3]. Foodborne disease is representative food safety issue, which results in social anxiety with economic loss of clinical and health costs [2,4,5]. Although most outbreaks are regarded as being linked to foods eaten outside the home, the private home has been reported as one of the major sources of foodborne illnesses [6,7,8,9]. Foodborne outbreak-associated illnesses have been reported as being attributed to foods consumed at home [8,10], and the number of actual cases is likely to be much higher than estimated, because numerous diseases with mild symptoms might be unreported [6,11].

Because consumers’ improper behaviors from the first step of the food preparation affect the risk level at following stages (e.g., food handling, cooking, and storage) [12], food purchase/transport should be regarded as the primary determinant factors for healthy food consumption. The potential risks of consumers’ food handling practices prior to meal preparation in the home can be represented by the growth of pathogens due to inadequate time-temperature control during food purchase and/or transportation [13,14,15]. Thus, major institutions that are responsible for food safety have provided the following consumer guidelines to support the proper behaviors during food purchase/transport: (1) keep food products out of the danger zone (4.4–60.0 °C; 40–140 °F) [16] by following the “2-h rule and 1-h rule” [i.e., preventing the exposure of food products for longer than 2 h if the temperature is (4.4–32.2 °C; 40–90 °F) and for longer than 1 h if the temperature is above 32.2 °C (90 °F)]; (2) place frozen foods and perishables (e.g., meats, poultries, fish, and eggs) in the shopping cart last; and, (3) refrigerate perishable products as soon as possible after purchase and preferably transport them with cold sources (e.g., freezer packs or ice) [14,15,17]. However, there is a lack of empirical data for the identification of consumers’ risky behaviors even though understanding consumers’ behaviors can contribute to the identification of critical control points where underrecognized microbiological hazards occur during food handling [18].

The present study aims to conduct consumer surveys on risky behaviors during food purchase/transport and their changes over time. Comprehensive analysis on time-temperature control is needed in order to estimate the actual level of risks derived from consumers’ food purchase/transport behaviors, as follows: (1) consumers’ preferences for food purchase and transport methods, including where and how temperature abuse can occur; (2) food purchase and transport time, including whether the exposure of food to ambient environments exceeds the recommended limit; and, (3) risk perceptions of food purchasing, namely, whether consumers aware of the proper purchasing order and which factors affect the purchased foods. However, there has been no survey research on food purchase and/or transport to analyze not only consumers’ preferences, but also the time that is required for each step and consumers’ risk perceptions regarding time-temperature control and microbiological risk factors [18,19,20]. Moreover, study designs of researches on consumers’ food purchase/transport behaviors as the determinants of the level of food safety risks were mainly based on the cross-sectional approaches without considering the changes in time frameworks [2,6,18,19,20,21]. We hypothesized that recent food shopping trends, especially for the changes in food retail formats and infrastructure of markets (e.g., high market share of large discount stores or online food shopping, traditional market decline, etc.) [22,23,24,25,26], have affected not only the consumers’ preferences for food purchase channels, but also their behaviors on food purchase/transport. Thus, we also expected that unchanged and/or emerging risky behaviors could be identified by the comparative analysis of two individual nation-wide surveys with same questionnaires over a decade [27]. We chose the longitudinal trend research approach for the analysis of distinct behavior changes. This novel approach is expected to overcome the major limitations of previous relevant researches that are associated with the topics of the present study (i.e., the link between food preparation and human health, food safety issues derived from consumers’ behaviors, researches on food purchase/transport, risk factors during food purchase/transport). In the case of the link between food preparation and human health, previous studies have mainly analyzed consumers’ behaviors with perspectives to the nutritional values (e.g., consumption of healthy foods, determinant factors on diet-related diseases, etc.) [28,29,30], whereas issues regarding the food safety risks (e.g., time-temperature control during food purchase/transport from the grocery store to the storage in household, etc.) were rarely reported. Moreover, researches on consumer behaviors for home food safety have mainly focused on hygienic practices and risk perceptions that are required in the kitchen (e.g., washing and trimming raw materials [31,32], cooking or serving foods [33,34], management of leftovers [35], etc.) [6,21], rather than food handling prior to the storage in household (i.e., food purchase/transport). Although microbiological risk factors that are linked to consumer behaviors in food purchasing to storage are regarded as crucial causes for foodborne diseases [12,36], the majority of the researches on food purchase/transport have focused on the understanding the motivation and intentions of food choice [37,38,39,40] without consideration of potential food safety risks.

In this study, quantitative surveys of primary food handlers were conducted in 2019 (*n* = 605) and 2010 (*n* = 609) with the same questionnaires regarding the following determinant factors for risky behaviors during food purchase/transport: consumer preferences for purchase/transport methods, time that is required for purchasing/transporting food products, and risk perceptions. A comparative analysis of the surveys was conducted to identify the major changes in each factor over time. Results from longitudinal analysis are expected to reveal the changes in the consumers’ behaviors during a decade and the insights whether those changes are positive or negative with the view to the proper consumer guidelines. The identification of emerging behaviors that can raise the risk level will imply the direction for establishing feasible countermeasures taken into consideration to drive the alteration of the behaviors of contemporary food consumers.

## 2. Materials and Methods

### 2.1. Participants

Adult consumers (i.e., people > 18 years old from South Korea) who were the main people involved in food preparation at home (hereafter defined as primary home food handlers) were recruited as the interviewees. Previous studies regarding home food safety have reported that hygienic perceptions and behaviors vary according to sociodemographic characteristics (gender, age, location, level of education, and family member) [8,41]; these characteristics were considered in the recruitment of participants. The participants (609 and 605 primary home food handlers in 2010 and 2019, respectively) were preallocated to stages by a multistage stratified systematic sampling method according to the statistical yearbook of population data in South Korea [42]. The sampling fraction used for geographic location was proportional to the total population. A survey was conducted in Various locations throughout Korea, including large cities (Busan, Daegu, Daejeon, Gwangju, Incheon, Seoul, and Ulsan), small/medium cities, and country towns. According to the survey methods for the longitudinal trend study over a decade, the participants were randomly recruited nationwide with homogeneity of sociodemographic characteristics both within and between surveys (survey 1 in 2010, survey 2 in 2019) to collect comparable responses. Table 1 shows the demographic characteristics of the participants. The participants were asked to answer the questionnaires after carefully thinking about their food purchase and transportation behaviors.

### 2.2. Development and Application of the Questionnaire

#### 2.2.1. Development of the Questionnaire

A questionnaire was used as the research instrument for the survey and a same questionnaire was used as the research instrument for the longitudinal trend surveys that were conducted in 2010 and 2019. A consulting committee organized by a government institution (Ministry of Food and Drug Safety, Cheongju-si, Korea) with experts in consumer surveys (Gallup Korea, Seoul, Korea), consumer organization, and food safety and hygiene laboratory (Korea University, Seoul, Korea) developed the questionnaire. All members of the consulting committee participated in composing a draft questionnaire and then revising the final version to verify its applicability. In addition, Gallup Korea reviewed the survey instrument for clarity and validity.

The questionnaires were developed for the identification of consumers’ behaviors with perceptions linked to the risks of food purchase/transport. Those behaviors and the questionnaire contents were mainly organized based on the internationally recognized food safety guidelines provided by major institutions, including Food and Drug Administration (FDA), Food Safety and Inspection Service in U.S. Department of Agriculture (USDA FSIS) [14,15,17]. Guideline-based information using for each major category of the questionnaire are as follows: (1) consumers’ preferences: major places which temperature abuse can occur during the food purchase/transport (i.e., purchasing channels and transport methods) and recommendation of proper handling of perishable products, (2) time-use for food purchase/transport: recommendation of time-temperature control represented as “2-h rule and 1-h rule” [i.e., keep foods under danger zone (4.4–60.0 °C) within 2 h (4.4–32.2 °C) or 1 h at >32.2 °C], and (3) risk perceptions: recommendation of food purchase order, including frozen/perishable foods and the importance on the information of food products. Firstly, the preferred methods of food purchase and transport were asked to respondents by providing choices of major purchasing channels and transport methods. Secondly, questions regarding the time respectively spent for the food purchase and transport were developed with the view to the time-use survey to estimate the food exposure time for ambient environments. Thirdly, questionnaires for consumers’ perception were designed to obtain information as to whether they were aware of the proper purchasing order, and which factors affected food purchases by investigating the consumers’ interests in essential information on products during the food purchase.

#### 2.2.2. Contents of the Final, Revised Questionnaire

To revise the draft questionnaire for clarity and validity, a pilot test was conducted by the pretesters who were consumers (*n* = 15; randomly selected) and expert researchers (*n* = 15; randomly selected). The pretesters were asked to evaluate the questionnaire with the perspectives to the terminologies that needed to be revised to improve clarity, unclear, and/or difficult expressions, and contents that might induce the respondents to feel displeasure or resistance during the survey [43]. The questionnaire was revised according to the opinions of the pretesters.

The developed questionnaire contained a total of nine questions on consumers’ food purchase and transport behaviors, including consumers’ preferences (Q1–Q2), food purchase/transport time (Q3–Q4), and risk perceptions (Q5–Q7). The specific questions were, as follows: Q1. Where do you buy each food (meats, fish/shellfish, fruit/vegetables, frozen processed foods, eggs, and others)? Q2. How do you transport purchased food from each place to your home? Q3. How long does it take you to buy food at each place? Q4. How long does it take you to transport the food? Q5. What do you buy first between food and nonfood items? Q6. What do you buy first between refrigerated/frozen foods or foods that can be stored at room temperature? Q7. Which factors do you consider when purchasing food?

### 2.3. Survey and Data Analysis

The present study is constructed with two individual surveys conducted in 2010 (survey 1; *n* = 609) and 2019 (survey 2; *n* = 605). The, same questionnaires were and method were used for sociodemographic group from each survey to examine how behaviors with risk perceptions from socio-demographic groups of primary food handlers have changed over time [44].

Face-to-face interviews with all respondents were conducted by trained interviewers from Gallup Korea at households or shopping centers. The instructions regarding the purpose of the survey were described at the top of the questionnaire, and the investigator briefly explained the background of the study prior to the interview. The results collected from each survey data were utilized for the comparative analysis to explore the key emerging changes in the consumers’ behaviors and/or perceptions during a decade.

All of the questions and responses were coded through the assignment of a unique number using the sui generis data coding system of Gallup Korea. The codes were entered in multivariate Excel spreadsheets. SPSS (Statistical Package for the Social Sciences, version 12.0, SSPS Inc., Chicago, IL, USA) was utilized to analyze the obtained data.

## 3. Results

### 3.1. Consumer Food Purchase Based on Establishment

#### 3.1.1. Changes in Consumers’ Preferences for Food Purchasing

Figure 1 shows the responses regarding the places where consumers preferred to purchase each food product category. Large discount stores were the most preferred places to purchase food in both surveys and they were ranked first for all food types in survey 2 (2019). Supermarkets near home and department stores showed little difference between surveys. However, decreases in consumers’ preferences for traditional markets and small markets near home resulted in changes in the rank orders, as observed by the results for frozen processed foods (from #3 to #4 and from #3 to #5, respectively). A considerable increase in the market share of home shopping, including online shopping, was apparent, especially for frozen processed foods (from 1.1% to 29.1%), fish/shellfish (from 1.1% to 10.9%), and other foods (from 1.2% to 20.3%). The changes in the responses from survey 1 (2010) to survey 2 (2019) were also distinct according to the purchase place, even though the level of changes was variable (Figure 1): (1) increases were found for large discount stores, supermarkets near home (except for meat, with a decrease of 0.9%), and home shopping, including online shopping; (2) decreases were found for small markets near home, traditional markets, and department stores.

#### 3.1.2. Changes in Consumers’ Preferences for Food Transport

Consumers’ preferences regarding transport methods also showed distinct changes toward more convenient methods in survey 2 (2019) when compared with survey 1 (2010): (1) an increase in the use of cars or delivery rather than walking to traditional markets or supermarkets/small markets near home, and (2) an increase in the use of delivery from large discount stores and department stores (Table 2). Growing preferences for using cars and delivery also resulted in a decrease in consumers transporting food by walking, even for traditional markets (73.1% in survey 1, 50.8% in survey 2). Most of the consumers from both survey 1 (2010) and survey 2 (2019) transported food using a car (in the trunk or on a seat in the car), especially for large discount stores (75.6 and 65.2%, respectively) and department stores (73.9 and 62.7%, respectively).

### 3.2. Total Food Purchase and Transport Time

Table 3 shows the average amounts of time required for food purchase and transport and the changes in these amounts of time between surveys. The participants in both surveys reported large amounts of purchase time for large discount stores in both surveys (survey 1: 79.1 min; survey 2: 76.5 min). The transport time for large discount stores and department stores increased in 2019 (survey 2), reaching 35.8 and 49.2 min, respectively (Table 3). Traditional markets and supermarkets/small markets near home also showed increases in both food purchase and transport time.

### 3.3. Food Purchase Behavior Based on Safety Guidelines

#### 3.3.1. Food Purchase Order

Consumers should buy nonfood stuffs and foods that can be stored at room temperature earlier, followed by refrigerated and frozen foods. However, as shown in Figure 2, our results revealed that many consumers purchased food in an improper order. In 2010, 35.5% of consumers bought food in the proper order (nonfood stuffs followed by food). Almost half of the consumers (46.0%) followed no particular purchasing order, and 18.6% of consumers even bought food in an improper order (buying food items before nonfood items). Over the nine years between the surveys, the number of consumers who bought the food in an improper order increased. A higher proportion of consumers (40.5%) bought food items first followed by nonfood items in survey 2 than in survey 1. In survey 2, 33.7% of consumers bought items in no particular order, and only 25.8% of respondents bought food in the proper order. When they were asked about their purchasing habits for refrigerated and frozen foods and foods that can be stored at room temperature, 47.6% of consumers followed the proper order (foods that can be stored at room temperature → refrigerated or frozen foods) in 2010. Approximately half of the respondents (47.1%) purchased food in no particular order, and some of the respondents (5.1%) purchased frozen and refrigerated foods before foods that can be stored at room temperature. In 2019, the majority (68.1%) bought food in the proper order, but the number of respondents purchasing refrigerated and frozen foods first before other foods increased (6.9%).

#### 3.3.2. Factors or Attributes Influencing Food Purchase

Respondents from both surveys indicated the ‘shelf life’ as the information of highest interest, as shown in Table 4 (92.9% in 2010, 92.1% in 2019). Most of the results, including the number of responses for each factor and the rankings, showed little change between surveys, except for nutritional information (i.e., nutritive components and calories). Distinct changes from survey 1 (2010) to survey 2 (2019) were the increase in the number of responses for ‘nutritive components’ (from 9.2% to 22.6%) and ‘calories’ (from 5.6% to 15.5%). Whereas, the interests in the ‘country of origin’ decreased in survey 2 (55.7% from 71.1% in survey 1). The other factors (e.g., ‘organic or not’, ’MSG added or not’, ‘transfat’, ‘other’, ‘none’, ‘no response’) did not show noticeable changes in the responses between two surveys.

## 4. Discussion

Problems in time-temperature control during consumers’ food purchase and transport have also been linked to the potential risks in food quality and safety, as the occurrence of the deterioration of food products at the post-harvest or post-processing levels are mostly attributed to inadequate infrastructure for storage and/or transport [45]. This trend survey provides practical information regarding determinant factors for risky behaviors during food purchase/transport: consumer preferences for purchase/transport methods, time required for purchasing/transporting food products, and risk perceptions. Based on the findings from the cross-sectional survey study (survey 1 conducted in 2010) highlighting the inappropriate perceptions/behaviors of food consumers corresponding to the food quality/safety issues, this longitudinal survey on a decade basis generally adopted by the trend study [27] was designed to establish the management strategies on ensuring the consumer food quality/safety by the clarification of whether those perceptions/behaviors are improved or not. A risky behavior reported in both surveys 1 (2010) and 2 (2019) was purchasing food in an improper order. In contrast, observations for survey 2 (2019) that showed changes from survey 1 (2010) were, as follows: (1) the market share shifted due to increased preferences for large discount stores and home/online shopping, (2) using cars and delivery were consumers’ preferred food transport methods, and (3) an unexpected increase in transport time, regardless of the purchasing channel resulting in long-term exposure of food to the ambient environment. Thus, purchase/transport time and purchase order were identified as the emerging and unchanged risk factors, respectively. The implications from these findings according to the major topics of this survey study can be summarized, as follows: food manufacturers and/or retailers should consider the changes in consumers’ preferences for food purchase/transport to prioritize more convenient methods and the unexpected increases in food purchase/transport time with improper food purchase order, which highlighted the underestimated risks in long-term exposure of foods under inadequate temperature.

Consumers’ preferences into purchase and transport channels implied the convenience as the noticeable trend [46,47]. The results on preferred food purchase channel likely follow the global grocery trends, which consistently report the drastic growths of large discount stores and online markets as modern food retailing methods [23,24], whereas the decline of traditional markets [26,48]. Online food delivery has also been regarded as the representative food purchase channel rapidly growing due to the convenience benefits [25]. The majority of the changes that emerged in survey 2 (2019) provide clues about risky situations that should be managed with adequate strategies for time-temperature control. Food that is selected by consumers is exposed to the ambient environment until it is transported to consumers’ home (except for home/online shopping) [2]. Consumer guidelines have stressed the time-temperature control of perishable foods, which should be stored at temperatures that are desirable for the prevention of microbial growth as soon as possible to avoid exposure to danger zones [6,15,17]. In particular, these guidelines emphasize time-temperature control in large discount stores, which induce consumers to purchase large quantities of both food and nonfood products; thus, relatively longer storage time at home is required [14]. However, distinct changes in consumers’ behaviors represented by the increase in preferences for large discount stores and the decrease in preferences for conventional purchasing channels that are suited to the purchase of small quantities of essential food products (small markets near home and traditional markets) suggest the importance of proper food purchasing habits. In the case of food transport method, using cars and delivery are convenient food transport methods; however, specific time-temperature control is also needed. Car trunks can cause rapid exposure to danger zones, particularly in sunlight [13,49]. In the case of grocery delivery, an increase in the market share of purchasing channels providing delivery services, especially for home/online shopping (as described above), is expected to enable the growth and popularization of delivery [25]. Although proper time-temperature control is needed during delivery by workers as well as during storage by consumers, there is a lack of background information regarding the risk factors in each step of food delivery. Temperature abuse can occur at any step prior to the storage of the delivered groceries, including the preparation, handling, delivery, and particularly the storage of products in the ambient environment of the final destination due to the delayed receipt of the groceries by the consumers [50,51]. Thus, practical consumer guidelines specialized for grocery delivery should be established based on the identification of control points in order to address those risk factors.

Increases in total food purchase and transport time suggest the importance of time-temperature control. Although food safety guidelines suggest following the “2-h rule and 1-h rule” (i.e., limit exposure of food to the danger zone to 1–2 h to prevent pathogen growth and/or toxin production under the temperature that can cause the growth or survival of foodborne bacteria in foods) [14,15,17,52], these results highlight the probability of long-term exposure of food products to ambient environments during purchasing and following transportation steps. Because the actual time that food is exposed to danger zones is determined by time-temperature control factors such as the purchasing order (i.e., buying perishable foods last) and the transportation environment (i.e., placing refrigerated foods in cooler bags with icepacks as a countermeasure for temperature abuse of perishable foods) [14,17,19], the importance of consumers’ proper behaviors should be stressed. However, risky behaviors have been consistently reported; for example, a consumer survey conducted by Karabudak, Bas, and Kiziltan [12] showed that only 4.8% of respondents transported raw meats in coolers after purchase. The increase in food transport time was unexpected because using cars was the dominant transport method based on the increasing preference for this method due to convenience, as described in Section 3.1 (i.e., the overall results suggest that convenience is one of the major causes for the changes in consumers’ preferences regarding food purchase and transport methods). Convenience is the major keyword in global food industry trends and it is generally exemplified in the decrease in time required for consumer food handling, especially for meal preparation time (i.e., time spent in home meal preparation and cooking) [18,53,54]; however, we revealed that food purchase and transport time was not affected by consumers’ preferences for convenience. The food purchase/transport time from major purchase channels has been rarely reported, and rather major researches regarding the time required for food preparation have focused on the meal preparation time [6,53]. Our findings can be a representative case for the analysis of food purchase/transport time according to the purchase place with the perspectives to the potential food safety risks. Further studies should be followed to reveal the reason and determinant factors of the purchase/transport time for the establishment of effective intervention strategies for time-temperature control.

Perceptions of microbiological risks that are derived from purchasing orders are representative unchanged risky behavior over time. These results indicate that many consumers overlook the importance of time-temperature control of food at the purchasing step. Although previous studies regarding consumers’ risky behaviors of food preparation have reported improved risk perceptions and knowledge (e.g., hygienic practices during the handling of raw materials and cooking, etc.) [21,55], improper purchasing order of perishable foods has been consistently reported [2]. Our results also indicate that consumers’ risk perception on food purchase order are required to be improved. Following an improper food purchasing order that does not provide appropriate time-temperature control can result in longer exposure of food to danger zones [14,15,17]; thus, the consideration of risk perceptions at the food purchasing step is necessary for the accurate assessment of the level of risk that corresponds to increases in total food purchase and transport time (as described in Section 3.2). In terms of factors influencing food purchase, consumer interest in the nutritional information of food products has increased. These results indicate global trends in food consumers’ interests shifting toward weight loss and increased awareness of healthy eating [56]. Trends for the consumption of healthy foods can induce increases in consumers’ preferences for purchasing nutritional and safe foods [47,57]. Information on nutritional values can be obtained by product labels (i.e., nutritive components and calories) and affect consumer interests, but, apart from shelf life, there is a lack of information regarding product safety during food purchases. Although both survey results showed that consumers were most interested in the ‘shelf life’ of foods purchased, risky behaviors were identified, as shown in Section 3.1, Section 3.2, and Section 3.3.1. To support consumers’ proper behaviors, our findings on such risky behaviors should be applied for the improvement of practical guidelines and the establishment of countermeasures.

This study newly identified risk factors of food purchase/transport, highlighting the impact of consumers’ behavior studies which have been mainly focused on hygienic practices during the food preparation steps after the food purchase/transport [21,58,59,60,61,62]. Our findings also implied the necessity for novel countermeasures and the improvement of current consumer guidelines against emerging (i.e., increased food purchase/transport time) and unchanged risky behaviors (i.e., food purchase order), respectively. Because the microbiological hazards that are derived from consumers’ risky behaviors are uncontrollable by national regulations, consumers’ proper risk perceptions, and knowledge are key prerequisites for the establishment of risk intervention strategies [8].

## 5. Conclusions

Our longitudinal trend study implied the necessity of the improvement on conventional consumer guidelines to cover contemporary trends in food purchase/transport with the perspectives to the time-temperature control during food purchase/transport: (1) chilling of perishable foods during transportation, especially in car trunks; (2) handling of delivered groceries from receipt to storage; (3) purchasing of food in the proper order; and, (4) compliance with the “2-h rule and 1-h rule” by considering the temperature of the environment to which foods are exposed and the total time required for food purchase/transport. Since unchanged and emerging risky behaviors observed in this study also highlighted the limitation of current risk management represented by the guideline, the development of novel consumer education materials and programs (e.g., infographics, leaflets, broadcasts, lectures, and videos) should be considered. To sum up, the implications from this research are expected to suggest food purchase/transport as underestimated topic in the research area of the link between food preparation behaviors and consumers’ health. Because this study focused on the identification of risky behaviors rather than the establishments of the causes and/or factors for those behaviors, further researches should be followed in order to identify the major determinants of identified risks (e.g., society psychological factors, environment factors, etc.), especially for the multiple factors that can increase risk levels.

## Figures and Tables

**Figure 1 ijerph-17-05448-f001:**
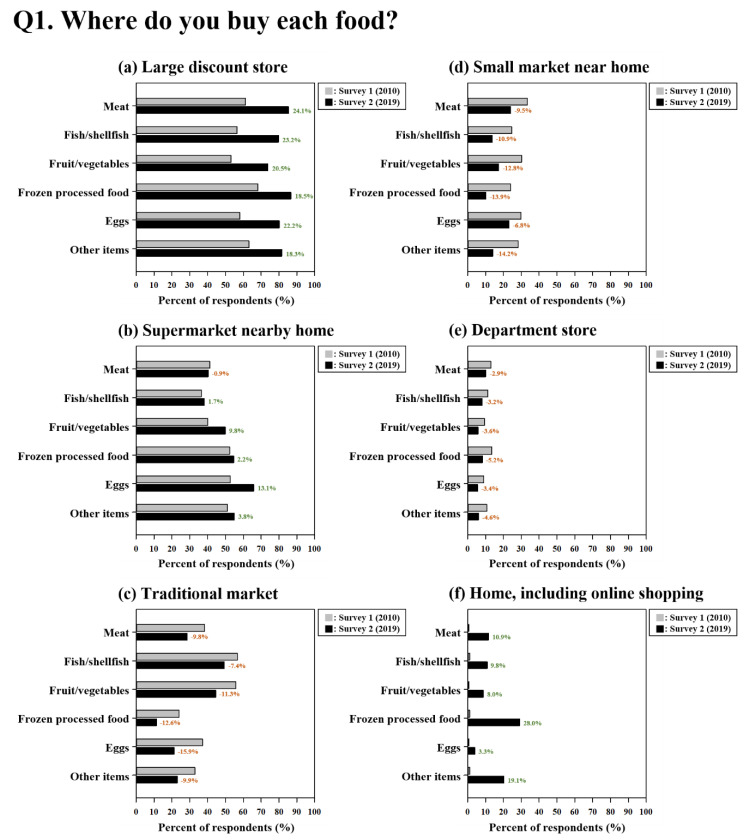
Purchasing places for meat, fish/shellfish, fruit/vegetables, frozen processed foods, eggs, and other items: (**a**) Large discount store, (**b**) Supermarket nearby home, (**c**) Traditional market, (**d**) Small market near home, (**e**) Department store, (**f**) Home, including online shopping.

**Figure 2 ijerph-17-05448-f002:**
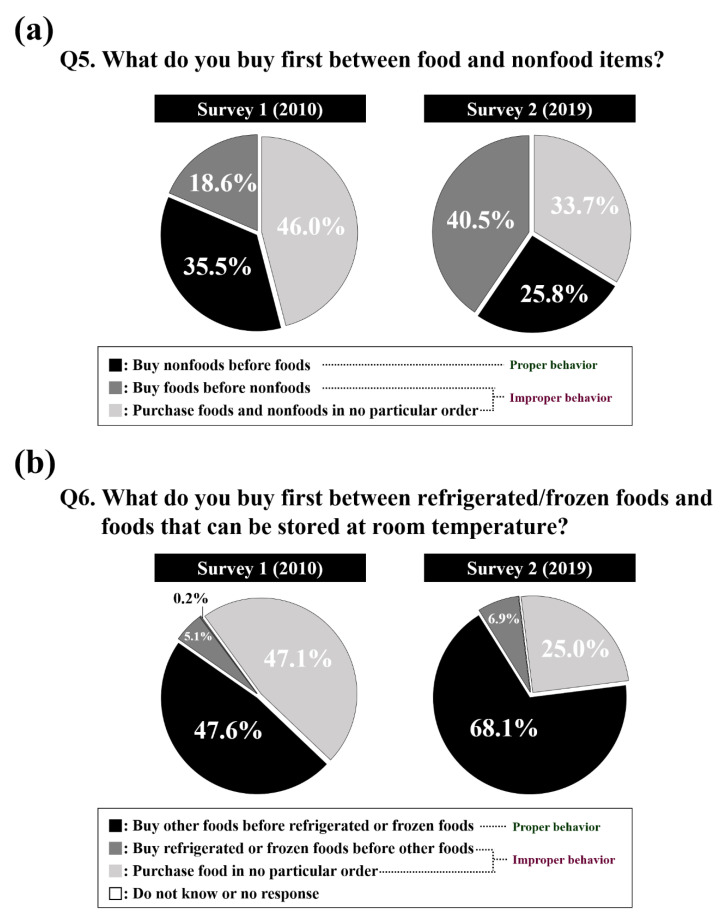
Consumers’ risk perceptions of the proper food purchasing order: (**a**) foods and nonfoods (**b**) refrigerated/frozen foods and foods that can be stored at room temperature.

**Table 1 ijerph-17-05448-t001:** Sociodemographic characteristics of the respondents.

Variables	Number of Respondents to Survey 1 (*n* = 609)	Number of Respondents to Survey 2 (*n* = 605)
Gender		
Male	49	48
Female	560	557
Age (years)		
19–29	67	74
30–39	144	144
40–49	148	145
50–59	112	112
>60	138	130
Location		
Large city	270	269
Small or medium city	271	269
Country town	68	67
Level of education		
Less than high school	122	120
High school	231	231
University	254	254
No response	2	-
Number of family members		
One person	64	47
2–3 persons	246	267
4–5 persons	266	257
More than six persons	33	34

**Table 2 ijerph-17-05448-t002:** Methods for transporting food from each purchase place to consumers’ homes.

Q2. How Do You Transport Purchased Food from Each Place to Your Home?							
Survey	Purchase Place	Percent of Respondents (Rank) ^1^					
		Walking	Delivery	Car Trunk	On a Car Seat	Other	Do Not Know/No Response
Survey 1 performed in 2010	Large discount store	21.4% (#3)	2.1% (#4)	50.3% (#1)	25.3% (#2)	0.8% (#5)	– ^2^ (#6)
	Traditional market	73.1% (#1)	0.5% (#5)	12.3% (#2)	12.3% (#2)	1.6% (#4)	0.2% (#6)
	Department store	19.8% (#3)	4.7% (#5)	50.6% (#1)	23.3% (#2)	1.2% (#4)	0.6% (#6)
	Supermarket or small market near home	90.5% (#1)	2.2% (#4)	2.9% (#2)	2.7% (#3)	– (#6)	1.6% (#5)
Survey 2 performed in 2019	Large discount store	17.9% (#3)	15.4% (#4)	44.3% (#1)	20.9% (#2)	1.6% (#5)	– (#6)
	Traditional market	50.8% (#1)	2.9% (#4)	25.3% (#2)	18.2% (#3)	2.9% (#4)	– (#6)
	Department store	22.6% (#3)	12.6% (#4)	39.0% (#1)	23.7% (#2)	2.1% (#5)	– (#6)
	Supermarket or small market near home	80.9% (#1)	4.6% (#4)	6.9% (#3)	7.1% (#2)	0.5% (#5)	– (#6)

^1^ respondents were asked to answer the most preferred method for transporting food from each purchase place. Percent of respondents and rank were calculated by the survey data within the same purchase place. ^2^ no one selected the option.

**Table 3 ijerph-17-05448-t003:** Food purchase and transport time.

Questions	Survey	Purchase Place	Average Time (min)
Q3. How long does it take you to buy food at each place?	Survey 1 performed in 2010	Large discount store	79.1
		Traditional market	45.4
		Department store	82.6
		Supermarket or small market near home	20.0
	Survey 2 performed in 2019	Large discount store	76.5
		Traditional market	56.2
		Department store	54.5
		Supermarket or small market near home	24.0
Q4. How long does it take you to transport food to your home?	Survey 1 performed in 2010	Large discount store	19.5
		Traditional market	16.4
		Department store	28.1
		Supermarket or small market near home	8.9
	Survey 2 performed in 2019	Large discount store	35.8
		Traditional market	30.8
		Department store	49.2
		Supermarket or small market near home	15.8

**Table 4 ijerph-17-05448-t004:** Consumers’ interest in major factors affecting food purchasing.

Questions and Choices	Percent of Respondents or Answers	
	Survey 1 (*n* = 609)	Survey 2 (*n* = 605)
Q7. Which factors do you consider when purchasing food? (multiple choices)		
Shelf life	92.9%	92.1%
Country of origin	71.1%	55.7%
Nutritive components	9.2%	22.6%
Organic or not	8.0%	5.0%
Calories	5.6%	15.5%
MSG added or not	3.6%	4.3%
Trans fat	1.8%	2.3%
Other	0.8%	2.5%
None	0.5%	–
No response	0.2%	–

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
