# Peer review of "Changes in Consumers’ Food Purchase and Transport Behaviors over a Decade (2010 to 2019) Following Health and Convenience Food Trends"

_ijerph, 2020, doi:10.3390/ijerph17155448_

Round 1
Reviewer 1 Report
The paper addresses an interesting topic on examining the changes in consumer food purchase and transport behavior related to health and convenience food trends. The authors use panel data between two periods (2010 – 2019) and a questionnaire to capture consumer responses. I think the authors have interesting data, but should structure these more around their underlying theoretical reasons for selecting this particular research. In other words, bring forward their theoretical contribution to the field in terms of clearly stating what they do and why they do it. Please see my comments below. I hope it contributes to guide you through your revision.
I think that the ABSTRACT is too wordy and it is specified more on demonstrating WHAT has been found rather than first showing WHY is this study important to consider. The abstract should be precise and clearly written in a way all readers will understand in a glance the importance of the research, the methodology used, the results, and the implications of the research, without reading the entire manuscript. Giving too many details on the results and only confused me a lot.
The INTRODUCTION is confusing. It is now clear what the authors aim to examine, the reasons that they aim in developing this research and the expectations of it. Since they do a comparison between 2010 and 2019 I was also expecting to find some literature on what might affect the changes (infrastructure, higher presence of supermarkets, higher presence of online food shopping etc.). Stating some hypothesis will also be useful and guide readers towards understanding the research.
Page 2, line 36-37 is very confusing. Explain what do you mean by convenient (I guess it is the type of grocery store [Supermarket, traditional markets, online food purchase]), and what do you mean by healthy food (I guess this terms is related to the nutrients and the perishability of the food), but then you mixed it with the obesity and foodborne disease. Then, what are the recent trends that you are referring but not mentioning here?
Page 2, line 45: I would disagree with you that safety issues related to food products are rarely reported, but I guess again that you are referring to the safety issues and the risks related with the food transportation from the grocery store to the household, distance, storage, food preparation etc. Please clarify these aspects.
Page 2, line 49 to 59: Is there any particular reason why you are focusing on foodborne disease?
Page 2, line 62 to 67: I think there is a mistake in the “2-hour rule” and the “1-hour rule” in terms of danger zones. If I am not mistaken, the correct would be preventing exposure of food for longer than 2 hours for a danger zone between 4-32ºC or preventing exposure of no longer that 1 hour if the temperature is above 60ºC.
Page 3, Line 89 to 98: If I understood correctly, besides examining consumer preferences for food purchase/transport methods, time required for purchasing/transporting food products, and risk perceptions, you also measure the efficiency of the official consumer guidelines related to food safety? In addition, I still struggle to understand how you measure consumers’ risk perceptions. Please explain.
The MATERIALS AND METHOD section needs to be completed. I would separate the survey methods from the participants, and then begin this section with the participants of your research. I would also like to congratulate the authors for the large and well balanced sample sizes between the two periods. Are the sample size sociodemographic characteristics representative of South Koreas’ population?
Please remove section 2.2.2 Pilot test of the draft questionnaire and just mention it briefly in the beginning of section 2.2.3.
Page 5, lines 182 to 184 are a repetition of Page 3, line 112-114. Please delete the most convenient for you.
The section of RESULTS should mention only the results. I would remove the discussion part from this section and create a separate section to discuss the results.
Page 6, line 200-201 I would remove this title and name this section under “Consumer food purchase based on establishment”. Please format the charts so that people who are willing to print in black and white can see the differences too.
Looks from your results (Table 2) that traditional markets and supermarkets or small markets near home are the most popular grocery store establishments for your customers, and they do not change by time (2010 vs. 2019). Also the majority of people transports the purchased food walking. Is that because traditional markets are near too? Please explain.
Shorten title 3.2 is very long.
Page 9, line 266: Is the minimum temperature of the safety guidelines 4ºC or 5ºC please correct.
Results from table 3: I would consider these average shopping times high. However do not know how they are considered in South Korea. Are there any studies to back up the results?
An interesting comparison here would be to estimate if there is a correlation between the results of table 2 and the ones of table 3.
Page 9, line 286: The title is not correct and title 3.3.1 is not a title but a sentence. I would suggest “Food purchase behavior based on safety guidelines”.
Please format the charts so that people who are willing to print in black and white can see the differences too.
Besides reporting frequencies another interesting results would be to examine whether there is a correlation between Q5 and/or Q6 and the type of establishments. For example, supermarkets normally are more organized, hence, it is easier for people to follow guidelines in comparison to traditional markets.
Page 11, line 316: please shorten the title. Also you are not including only the nutritional information but other product attributes as well. Hence, I would either entitle this part as “Factors or attributes influencing food purchase”. Again, besides presenting frequencies, it would be interesting and informative for the industry and policy makers of healthy food what are the most important factors that South Korean consumers consider when purchasing food. This can be obtained by estimating a factor analysis.
The section of DISCUSSION should discuss the results compared with other studies. I highly suggest the author(s) to remove the discussion and the literature from the results and place it in this section. Discussions should also be framed around the main objective of the research.
Finally the CONCLUSION section should include specific messages for the manufacturers, the industry and the policy makers since you are measuring the efficiency of food safety guidelines. The conclusions are very general and should provide a clear message on what needs to be changed based on the results of this paper, what possible strategies might be used to solve the problem, and how can the manufacturers and/or the public bodies intervene to solve part of the problem.
Reviewer 2 Report
Dear Authors:
Your manuscript entitled "Changes in Consumers’ Food Purchase and Transport Behaviors Over a Decade (2010 to 2019) Following Health and Convenience Food Trends ", is not well defined and should better explained.
Comments:
Abstract: The abstract is too long and there should more clearly stated the main aims, possible novelties and/or contributions, main few findings, and implications and not only the results of the two surveys (2010 and 2019).
Structure of the manuscript:
The manuscript is not well structured. It needs more clear structure on introduction, review of literature, data and methods, empirical results and discussion, implications, and conclusion.
In addition, the manuscript needs several clarifications and improvements:
1. Introduction:
The section on the Introduction is without clear motivation and without a focus. It is suggested to be divided into three parts: first, introduction, where is used less literature, but more clearly defined motivation, aims and objectives, and possible novelty and/or contribution of the manuscript. Second, literature review, because there is missing a strong section on review of relevant literature. The listed references there are few.
Lines 58-59: to inadequate temperature-time control during food purchase and/or transportation. This is mostly attributed to poor agricultural practices, technical limitations, financial and labour restrictions, and inadequate infrastructure for storage, processing, and transport: see and cite the following work:
- Fanelli R.M. (2019) Using Causal Maps to analyse the major root causes of household food waste: Results of a survey among people from Central and Southern Italy. SUSTAINABILITY, 11(4), 1183; https://doi.org/10.3390/su11041183.
Clarifications: It is not clear about the selection of the analysed period (2010 and 2019). Please motivate your choice.
2. Materials and Methods:
The data from the questionnaires and supplementary documentation were analysed using just simple graphs and/or tables.
I suggest to use descriptive statistics (min, max, standard deviation, coefficient of variation) to explain the features of the two sample and linear regression to identify the main determinants of food preparation behaviors linked to consumers’ health. Especially the multiple factors that can increase consumers’ risky behaviour. Risky behavior could be used as a dependent variable in a linear regression model, with the other variables being inputted as predictors.
3. Results and Discussion
The results and findings should be discussed and compared with previous studies and findings.
What are the study implications?
4. Conclusions
Conclusions should be improved as they largely repeated the results.
What are the study limitations?
What are the proposals for research in future?
Tables and figures:
Regarding Tables and Figures:
Quality of Figures should be improved.
Round 2
Reviewer 1 Report
I would like to congratulate the authors. They made a very good job and high effort in alleviating all my concerns. In particular, the introduction now clearly states the aim of the paper and what is expected (hypothesis) from it. Results are coherent with a clear outcome. Yet, I have several minor comments that needs to be addressed. See comments bellow.
You could merge sections 2.3. Survey and 2.4. Data analysis under one section named “Survey and data analysis” to simplify things.
In section 3.3. write ‘behavior’ instead of ‘behaviour’
Page 11, lines 279-281: Please homogenize the aim of the paper as in the introduction. “…..determinant factors of food preparation behaviors….”.
Page 11, lines 293-297: Please briefly specify the main actors that will benefit from the implications of your results instead of numbering them. For example, food manufacturers and/or retailers should consider the changes in consumers’ preferences for food purchase/transport to prioritize more convenient methods (copied from your original manuscript).
Reviewer 2 Report
The Authors, based on my suggestions and observations, have enriched and improved the manuscript, therefore the same can be published in the present form.
Author Response
Thank you for the helpful comment.